# Isotopic and Geophysical Investigations of Groundwater in Laiyuan Basin, China

**DOI:** 10.3390/s24217001

**Published:** 2024-10-31

**Authors:** Weiqiang Wang, Zilong Meng, Chenglong Wang, Jianye Gui

**Affiliations:** 1Institute of Hydrogeology and Environmental Geology, Chinese Academy of Geological Sciences, Shijiazhuang 050061, China; agui_mail@163.com; 2Key Laboratory of Groundwater Sciences and Engineering, Ministry of Natural Resources, Shijiazhuang 050061, China; 3Hebei Institute of Hydrological Engineering Geology Exploration, Shijiazhuang 050021, China; mengzilong10@163.com (Z.M.); hbskywt@126.com (C.W.)

**Keywords:** Laiyuan basin, hydrogen and oxygen isotopes, magnetotelluric, impedance tensor decomposition, 2D inversion, groundwater

## Abstract

Due to the complex intersection and control of multiple structural systems, the hydrogeological conditions of the Laiyuan Basin in China are complex. The depth of research on the relationship between geological structure and groundwater migration needs to be improved. The supply relationship of each aquifer is still uncertain. This paper systematically conducts research on the characteristics of hydrogen and oxygen isotopes, and combines magnetotelluric impedance tensor decomposition and two-dimensional fine inversion technology to carry out fine exploration of the strata and structures in the Laiyuan Basin, as well as comprehensive characteristics of groundwater migration and replenishment. The results indicate the following: (i) The hydrogen and oxygen values all fall near the local meteoric water line, indicating that precipitation is the main groundwater recharge source. (ii) The excess deuterium decreased gradually from karst mountain to basin, and karst water and pore water experienced different flow processes. (iii) The structure characteristics of three main runoff channels are described by MT fine processing and inversion techniques. Finally, it is concluded that limestone water moved from the recharge to the discharge area, mixed with the deep dolomite water along the fault under the control of fault F2, and eventually rose to the surface of the unconsolidated sediment blocked by fault F1 to emerge into an ascending spring.

## 1. Introduction

With the continuous development of human society and economy, the demand for water resources is increasing. The shortage of water resources has become an important restrictive factor that hinders social development. Therefore, it is of great practical significance to study the circulation of groundwater recharge, runoff, and discharge for the utilization and protection of groundwater resources. At present, the environmental isotope method is the main method of groundwater circulation. Hydrogen and oxygen isotopes participate in various processes of water circulation and transformation, record key information about water transport processes, and solve many problems related to water circulation [1,2]. However, isotopes can only analyze the source and circulation conditions of water from a macroscopic perspective, and cannot accurately locate the runoff channels of groundwater. Geophysical methods can provide detailed information on underground lithology changes and hydrogeological structures. Electrical resistivity tomography (ERT), magnetotelluric (MT), electromagnetic probing, and gravity-based methods have been widely applied to the structural characteristics of unconsolidated sediment aquifers [3], detection of groundwater flow paths in Fractured Rock Aquifers [4,5], and groundwater flow in merokarst systems [6]. As of now, scholars have integrated the geophysical survey and hydrogeological chemical techniques to solve the problems of comprehensive identification and water source protection of karst groundwater systems [7,8], the relationship between groundwater chemical characteristics and lithology and geological structures [9,10], and groundwater potential evaluation [11,12,13].

The Laiyuan Basin in the Taihang Mountains of North China is a typical syncline karst basin with a severe water shortage. Spring groups developed in the basin. In recent years, with the decrease in precipitation and the increase in groundwater exploitation, as well as the transformation of the surrounding environment of the spring group by human activities, most of the spring group is drying up, resulting in a serious water shortage. Therefore, some scholars have studied the groundwater of the Laiyuan Basin. As per the latest changes in groundwater dynamics, it is indicated that the groundwater level in the Laiyuan Basin exhibits periodic variations. In high flow years, the precipitation is large, groundwater exploitation is small, the groundwater recharge amount is also large, and the groundwater level rises more. In dry years, the precipitation is small, the exploitation is large, and the groundwater level drops greatly. Overall, the groundwater is in a state of overexploitation for a long time, and the dynamics of the water level in the basin shows a slow declining trend. According to the latest hydrogeological characteristics research, field hydrogeological survey methods are mainly used to qualitatively analyze the controlling effect of geological structures such as faults on groundwater flow fields. Based on simulations of the groundwater chemical evolution in the basin, the karst water supply to pore water and the effect of them mixing are qualitatively analyzed, but the specific runoff path of the karst water supply to pore water is not identified. At the same time, hydrogen and stable oxygen isotopes are used to qualitatively reveal the recharge, runoff, and discharge processes of each water-bearing rock group in Laiyuan Basin, but the runoff channels of each water-bearing rock group cannot be verified. Geological structure plays an important role in the structure and flow field of karst groundwater [14,15]. At present, the research depth of the relationship between geological structure and hydrogeology in this area needs to be improved. The runoff channel of each aquifer formation is directly related to the formation structure and fault structure, so it is necessary to combine geophysical techniques for fine localization. Commonly used hydro-geophysical methods such as ERT, electromagnetic probing, the radar method and others can be used; for these, exploration depth is shallow, generally only for shallow groundwater structure detection. The MT method is a natural-source electromagnetic wave method with the characteristics of large exploration depth and high shallow resolution. By observing the electromagnetic field intensity of electromagnetic waves propagating through different geological bodies, the underground resistivity can be inferred to provide information on geological structure characteristics, aquifer structure, rocks’ physical parameters, etc., which are applicable to different terrain conditions [16,17].

This paper combines geochemical and geophysical methods. Based on the characteristics of hydrogen and oxygen isotopes, the sources and mixing of aquifer formations in different hydrogeological units are revealed from a macroscopic perspective. The characteristics of the tectonic structure controlling groundwater migration in the Laiyuan Basin are characterized comprehensively by using MT processing and inversion technology, and the formation structure and fracture and karst fissure development law for each water-bearing rock group are investigated in detail. The two methods support each other, from macro-analysis to micro-channel comprehensive characterization of groundwater recharge, runoff, and discharge, which is of great significance for the rational development, utilization and protection of groundwater resources in areas with severe water shortages.

## 2. Geological and Hydrogeological Settings

The Laiyuan basin is located in the northern section of Taihang Mountain in China; the terrain is generally high in the northwest and low in the southeast. Figure 1 shows the hydrogeological features of the Laiyuan basin. The structural features within the area are mainly characterized by open and gentle folds and a northeast-oriented fault structure pattern. According to geological conditions and geomorphic features, the study area is divided by the high mountain in the northwest, the low mountain in the southeast, the basin, and the river. The northwest high mountain belongs to the Lai-yi anticlinal sedimentary rock area, mainly exposed to Proterozoic dolomite and Cambrian, Ordovician limestone, in which the Tuyu syncline is distributed in the northeast direction, and the hinge of the syncline is located in the Hon-Zgm-Shipi area. The southeastern low mountain area belongs to the Shanxi platform anticline volcanic area, and the Laiyuan basin is covered by unconsolidated sediment.

The Laiyuan Basin is a closed basin in which the surface watershed is basically the same as the underground watershed. The watershed in the southeast is an intrusive rock mass and upper Proterozoic gneiss aquifer containing weathered fissure water. In the tuyu syncline in the northwest, groundwater flows to the core, and the surface watershed tends to coincide with the underground watershed. The northeast is the upwelling end of the tuyu syncline, the overflow recharge is small, and the surface watershed and underground watershed tend to be the same. The end of the tuyu syncline is in the southwest, and the watershed is located in the core of the syncline; the groundwater level is higher here, and the surface watershed is basically the same as the underground watershed.

Precipitation is the main recharge source of groundwater. In the karst mountain area outside the basin, some of the precipitation flows into the valley and gully of the low-lying mountain, and then flows downstream in the form of subsurface flow. The other part is directly converted into groundwater through vertical infiltration such as with solution gaps and solution holes, and then with downward runoff. During the runoff process, a small portion is exposed in the form of spring, while the majority continues to move towards deeper depths along the solution gap and solution hole. The groundwater in this area can be divided into four types: unconsolidated sediment pore water, carbonate rock karst water, tertiary pore water, and Jurassic volcanic fissure water.

## 3. Materials and Methods

### 3.1. Sampling and Analysis of Hydrogen and Oxygen Isotopes

When the water temperature is low, the effect of rocks’ isotopic compositions and temperature on groundwater isotopic fractionation can be ignored. The groundwater temperature in the Laiyuan area is low in winter and the temperature changes are small, of about 6~13 °C. As such, we sampled in winter 2020. Figure 1 shows 51 groups of groundwater samples which were collected in the carbonate rock karst water area, the Laiyuan Basin, and the gully valley area. The coordinates and elevation of the sampling points were measured by GPS. Samples were collected in 500 mL plastic bottles and stored at a low temperature. The test instrument was a MAT253 gas isotope mass spectrometer, the product of the Thermo company in the United States, equipped with Gasbench II and FlashEA1112HT (ConFlo VI interface). The δD was determined by high-temperature thermal conversion elite-isotope-ratio mass spectrometry (HTC-IRMS) with an accuracy of 1.0‰. δ^18^O was determined by Gasbench II isotope-ratio mass spectrometry (Gasbench II IRMS) with an accuracy of 0.2‰. The testing and analysis work were completed at the Key Laboratory of Groundwater Sciences and Engineering, Ministry of Natural Resources.

### 3.2. Collection and Processing of MT

MT is a passive electromagnetic (EM) method for probing the subsurface electrical resistivity structure of the Earth. The information on the subsurface electrical structure contained in MT data is derived from the simultaneous measurements of natural time-varying electric (E) and magnetic (H) fields in the orthogonal horizontal direction on Earth’s surface [18,19].

As shown in Figure 1, in the vertical geological tectonic unit direction of the study area (Tuanyuan syncline, Laiyuan basin, main karst fracture tectonic belt), three MT survey profiles are arranged along the direction of the groundwater flow field, with an effective frequency of 1000~0.35 Hz. Two Canadian Phoenix MTU-5A devices were used in the field. The single point acquisition time was more than 2 h, the effective acquisition frequency was 1000~0.35 Hz, and the interval between stations with MT data was approximately 250 m. Among them, 58 points are arranged on the MT01 profile, 56 on the MT02 profile, and 83 on the MT03 profile. To clearly display the map, the MT point number is marked at every 5 iterations. Mt sites should be kept away from areas with large electromagnetic interference as much as possible, and nighttime collection should be selected for areas with large amounts of human interference during the day.

After collecting MT data, we use SSMT2000 processing software to convert the collected raw time series of MT data into power spectrum data. The multi-site, multi-frequency tensor decomposition technology of MT is employed to decompose the rose diagram of geo-electrical strikes and the site-based cloud diagram for the identification of linear geological structures (MT-Pioneerversion:6.0). Additionally, the structures of aquifers are accurately imaged by two-dimensional inversion technology (Nonlinear conjugate inversion algorithm).

## 4. Results and Discussion

### 4.1. Characteristics of Hydrogen and Oxygen Isotopes

As shown in Figure 2, the δD-δ^18^O relationship was plotted from test results for 51 groups of samples; the selection of the local meteoric water line (LMWL) equation in China is δD = 7.9δ^18^O + 8.2 [20]. The δD and δ^18^O values of dolomite, limestone, and unconsolidated sediment aquifers all fall on or near the LMWL, which indicates that precipitation is the main recharge source for groundwater in the study area.

As shown in Figure 2, the δD-δ^18^O values in the Xie valley zone of the limestone aquifer are similar to those in the syncline core, indicating a recharge relationship between the two regions. It is speculated that during the process of karst water flowing from northeast to southwest in the core of the syncline, groundwater accumulates in the Xie Valley zone. The distribution of δD-δ^18^O values near Hon-Aih-Los is similar, and it is speculated that there is a runoff zone along this path.

As shown in Figure 2, the δD-δ^18^O values of unconsolidated sediment aquifer are distributed in the lower right corner of the LMWL, indicating that groundwater experienced evaporation before its formation. The overall distribution relationship of δD-δ^18^O values near Beh and Beg is similar, suggesting the existence of a runoff zone along the corresponding path. As shown in Figure 2 the overall distribution of δD-δ^18^O values in unconsolidated sediment aquifer tends to be that of limestone and dolomite. It is inferred that unconsolidated sediment aquifer not only receives precipitation, but also receives lateral recharge of limestone and dolomite karst water.

As shown in Figure 2, the δD-δ^18^O values of samples 2–40, 3–36, 3–38, 3–32, 3–34, 1–53 (Los-Xiz–Bpo-Jin) of the dolomite aquifer are located in the distribution area of the δD-δ^18^O values of the limestone aquifer, which suggests that the limestone aquifer mixed into the dolomite aquifer.

### 4.2. Characteristics of Deuterium Excess

The deuterium surplus value(d) reflects the degree of isotopic fractionation of precipitation during the evaporation and condensation process, defined as follows: d = δD − 8δ^18^O [21]. The influencing factors of deuterium surplus in the aquifer include the following: the properties and quantities of oxidizing chemical components in aquifers, the solubility of the rocks’ mineral crystal structure, the physical and chemical properties of groundwater, such as pH, temperature, and redox state, the open or closed state of the aquifer system, and the retention time of groundwater in aquifer [22]. The main factors that affect the variation in the d value within the same aquifer are roughly the same. The relative variation in the d value is mainly related to the length of time it stays in the aquifer. Groundwater runoff has a longer route, slower runoff speed, and longer retention time.

As shown in Figure 3a, for the dolomite aquifer, the d value of the valley zone from both sides of the reunion syncline to the periphery of the basin decreases, indicating the genetic characteristics of groundwater control by structure and geomorphology. On both sides of the syncline, the terrain slope is large and the formation dip angle is steep, which is conducive to groundwater runoff. The valley development zone is located in the periphery of the basin, with relatively small topographic relief, flat formation, a long groundwater runoff path, a slow flow rate, and long retention time.

As shown in Figure 3b, for the limestone aquifer, the d value decreases from the watershed boundary to the foothills, reflecting the overall water cycle characteristics of karst water from recharge to runoff and to the discharge zone. The recharge area and runoff area are karst mountainous areas with large hydraulic slopes and fast groundwater regeneration. When the discharge area is located in the foothills, the hydraulic slope decreases, the runoff speed slows, the geological environment is relatively closed, and the retention time becomes longer.

### 4.3. Characteristics of Magnetotelluric Field

Based on isotopic characteristics, it is speculated that there may be three runoff zones: Zgm-Shq-Xiz, Jin-Beh-Beg, and Hon-Aih-Los. As shown in Figure 1, three MT profiles are arranged along the direction of the runoff zone to finely depict the spatial distribution of karst, fractures, and structures in the runoff zone. The three MT profiles are named MT01, MT02, and MT03, respectively.

#### 4.3.1. Impedance Tensor Decomposition of MT

Impedance tensor decomposition technology can suppress the influence of three-dimensional small anomalies near the surface, providing reliable regional impedance tensor data for 2D inversion, and obtaining geo-electrical strikes with all sites and all frequencies [23]. After the collection of the field time series data, the power spectrum of the single measuring point is calculated through preliminary processing, then the conjugate impedance method (CCZ) decomposition is performed (MT-Pioneer software by Chen) [23]. The geoelectrical structure can be identified by analyzing the geo-electrical strikes and structural dimensions of the profile [24].

Figure 4 shows the rose diagram of the MT01 profile; the distribution of the geo-electrical strikes in this profile is obvious and the two-dimensional property is good, and the azimuth of the geo-electrical strikes in the first quadrant is mainly distributed in the range of 10°–60° (Table 1). Due to the complexity of the tuyu syncline structure, there are four geo-electrical strikes in Karst mountain (i), and the most significant ones are at NNE20°–41°. The geo-electrical strikes of the diluvial slope are at NNE30°. The geo-electrical strikes of the Laiyuan Basin (alluvial slope, river valley) have decreased to NNE15°.

Figure 5 shows the rose diagram of the MT02 profile; the distribution of the geo-electrical strikes in this profile is obvious and the two-dimensional property is good, and the azimuth of the geo-electrical strikes in the first quadrant is mainly distributed in the range of 5°–72° (Table 2). Due to the complexity of the tuyu syncline structure, there are three geo-electrical strikes in the Karst mountain (i), and the most significant ones are NNE41°–5°. The geo-electrical strikes of the diluvial slope are at NNE5°. There are two geo-electrical strikes in the Laiyuan Basin (alluvial slope, river valley), with NNE9° being the most significant. The geo-electrical strikes of the southernmost karst mountainous area on the profile are at NEE42°.

Figure 6 shows the rose diagram of MT03 profile; the distribution of the geo-electrical strikes in this profile is obvious and the two-dimensional property is good, and the azimuth of the geo-electrical strikes in the first quadrant is mainly distributed in the range of 5–30° (Table 3). Due to the complexity of the tuyu syncline structure, there are three geo-electrical strikes in the Karst mountain (i), and the most significant ones are at NNE28°. The geo-electrical strikes of the diluvial slope are at NNE5°. There are two geo-electrical strikes in the Laiyuan Basin (alluvial slope, river valley), with NNE9° being the most significant. The geo-electrical strikes of the easternmost karst mountainous area on the profile are at NEE19°.

To sum up, it can be inferred that the geo-electrical strikes of geologic bodies in the Laiyuan basin are mainly located at 0°–60°, within which NNE20~45° is the most significant in the karst mountain (unity syncline) area, NNE30° and NNE5° in the diluvia slope are, and NNE9° the in alluvial slope area. Therefore, the distribution difference in geo-electrical strikes of different geological and geomorphic bodies is obvious, and there are multiple two-dimensional electrical geological bodies in the tuyu syncline area of the karst mountains, which is presumed to be controlled by multiple faults.

#### 4.3.2. Two-Dimensional Inversion of MT Data

Impedance tensor decomposition is used to determine the strikes and the strong two-dimensional characteristics of the geological structure in the study area. On this basis, TE\TM polarization mode recognition is carried out. The statistical results show that the resistivity model based on TM polarization mode data has a higher resolution for geological structure and geologic body anomaly zones [25]. Therefore, in this paper, TM polarization data are rotated to the geo-electrical strikes for 2D inversion. The inversion algorithm adopts a two-dimensional nonlinear conjugate gradient method with topography. The one-dimensional Bostick inversion results of the TE model are used as the initial resistivity model. The inversion cell size increases in depth by 10 m. In the inversion process, the L curve analysis method is used to determine the value of the regularization factor 25 (MT-Pioneer software by Chen) [26].

The characteristics of stable isotopes and the deuterium surplus value qualitatively infer that the limestone aquifer on the MT01 line is controlled by the syncline structure. The karst water is collected in the Xie valley, and there is a karst runoff belt in Zgm-Shq-Xiz to the basin. As shown in Figure 7, in the tuyu syncline karst mountain area of the MT01 profile (Zgm-Shq-Xiz), the Cambrian limestone and the Quaternary gravel layer are characterized by medium and low resistance, which is inferred to be the groundwater runoff channel. The resistivity at fault F4, F6, and F7 shows high resistance characteristics, suggesting that karst fractures are not developed. The resistivity of the F5 fracture is low, and the karst fissure is developed. This indicates that karst water accumulates in the F5 fault, which is located near Zgm, which is consistent with the inferred results of isotope characteristics. The bottom of Cambrian strata is the water-resisting layer with shale, so there is little connection between Ordovician karst water and Mesoproterozoic dolomite karst water. At the F2 fault, the Cambrian strata began to loosen, and the limestone karst water without a water-resisting layer was merged into the tertiary sandstone aquifer and Mesoproterozoic dolomite aquifer along the fault F2. Therefore, the Mesoproterozoic karst water collects part of the Cambrian karst water along the F1 fault and eventually emerges from the surface as an ascending spring.

The characteristics of stable isotopes and the deuterium surplus value qualitatively inferred that the distribution of the δD-δ^18^O values near Jin-Beh-Beg in the MT03 line is similar. and it is speculated that there is a runoff zone along Jin-Beh-Beg. Affected by concealed fault zones, the karst water of dolomite is mixed with limestone water, and the loose aquifer is mixed with dolomite and limestone water. As shown in Figure 8, in the Jin-Beh-Beg area of MT02 profile, the Cambrian limestone and the Quaternary gravel layer are characterized by medium and low resistance, which is inferred to be the groundwater runoff channel. The bottom of the Cambrian strata is the water-resisting layer with shale, so there is little connection between Ordovician karst water and Mesoproterozoic dolomite karst water. The water-conducting buried fault F2 has allowed limestone water to be mixed into the tertiary sandstone aquifer and Mesoproterozoic dolomite aquifer.

The characteristics of the stable isotopes and the deuterium surplus value qualitatively inferred that the distribution of the δD-δ^18^O values near Hon-Aih-Los in the MT03 line is similar. And, it is speculated that there is a runoff zone along Hon-Aih-Los. Affected by concealed fault zones, the karst water of dolomite is mixed with limestone water, and the loose aquifer is mixed with dolomite and limestone water. As shown in Figure 9, in the tuyu syncline karst mountain area of the MT03 profile (Hon-Aih-Los), the Cambrian limestone and the Quaternary gravel layer are characterized by medium and low resistance, which is inferred to be the groundwater runoff channel. The bottom of Cambrian strata is the water-resisting layer with shale, so there is little connection between ordovician karst water and mesoproterozoic dolomite karst water. The development of the buried fault F2 in this section leads to the mixing of water bodies. Therefore, the geophysical results have described the aquifer communication, recharge, and discharge channels inferred by stable isotope characteristics in detail, and achieved qualitative and quantitative characterization.

## 5. Conclusions

This study on the hydrogeological structure of the Laiyuan Basin, using the characteristics of hydrogen and oxygen isotopes, impedance tensor decomposition, and 2D inversion of MT shows that the fault structure plays an important role in groundwater migration in this area, and the following conclusions are drawn about groundwater recharge, runoff, and discharge.

The distribution of hydrogen and oxygen isotopes indicates that atmospheric precipitation is the main source of groundwater in the study area.According to the distribution characteristics of hydrogen and oxygen isotopes and deuterium surplus in groundwater, three groundwater runoff zones, Zgm-Shq-Xiz, Jin-Beh-Beg, Hon-Aih-Los, were qualitatively identified. Karst water and unconsolidated sediment pore water have undergone different flow processes, which reveals the controlling effect of structural and geomorphic form on groundwater circulation.The geophysical results have described the aquifer communication, recharge, and discharge channels inferred by stable isotope characteristics in detail, and achieved qualitative and quantitative characterization.

In this paper, the geological structure of each aquifer in the Laiyuan Basin is described in detail, which is mutually confirmed with the characteristics of hydrogen and oxygen isotopes, which has important practical significance for the study of the characteristics of groundwater recharge and runoff. Based on these characteristics and the current situation of water shortages and dry springs, geophysical methods can be used to find water projects to solve short-term water shortage problems and alleviate the current water shortage situation. The local government has gradually restored the groundwater table and spring water by strictly controlling the groundwater exploitation in the relevant runoff zone for a long time. By limiting the discharge of polluted water from industry and agriculture, planning wastewater treatment projects, and controlling the elimination of man-made pollution sources, the sustainable management of groundwater quality is effectively ensured.

## Figures and Tables

**Figure 1 sensors-24-07001-f001:**
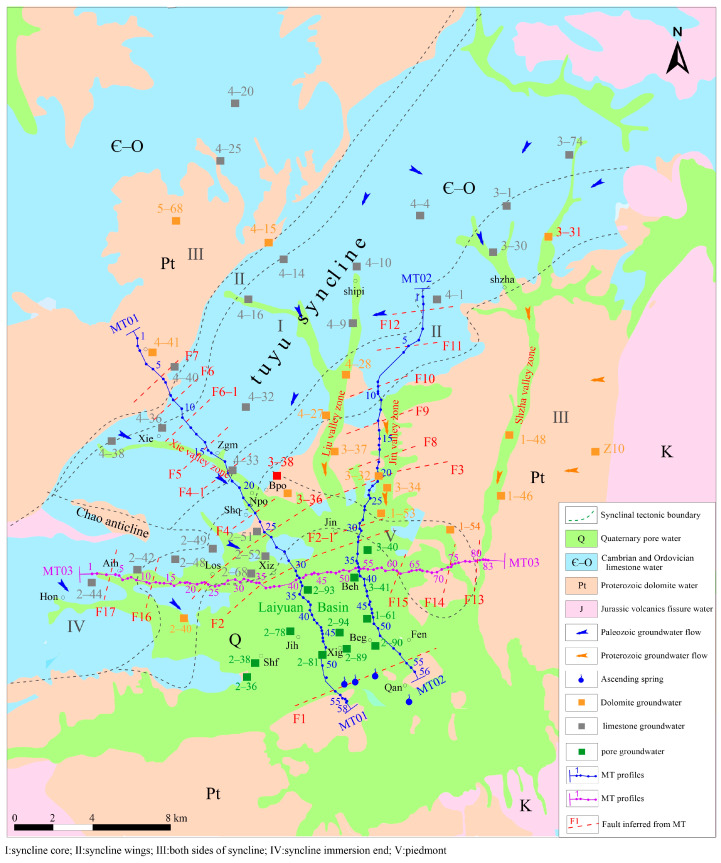
Location, sampling sites and hydrogeological map of the study area.

**Figure 2 sensors-24-07001-f002:**
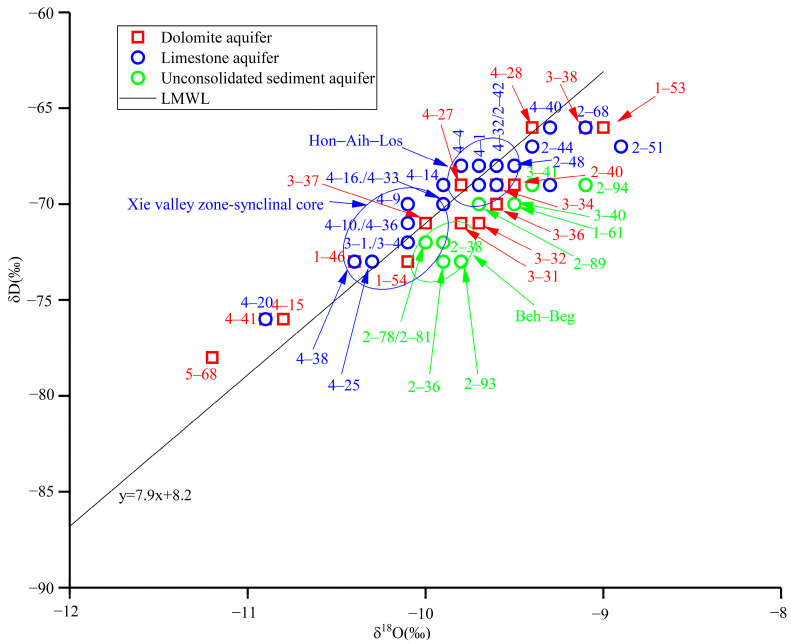
Characteristics of groundwater δD-δ^18^O in Laiyuan Basin, China.

**Figure 3 sensors-24-07001-f003:**
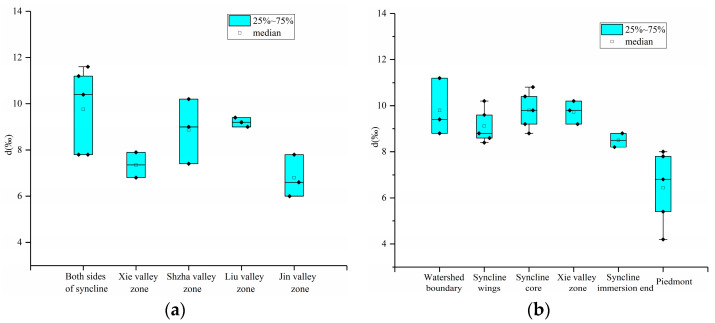
Characteristics of groundwater deuterium excess (d) in the Laiyuan Basin, China, (**a**) dolomite aquifer. (**b**) Limestone aquifer.

**Figure 4 sensors-24-07001-f004:**
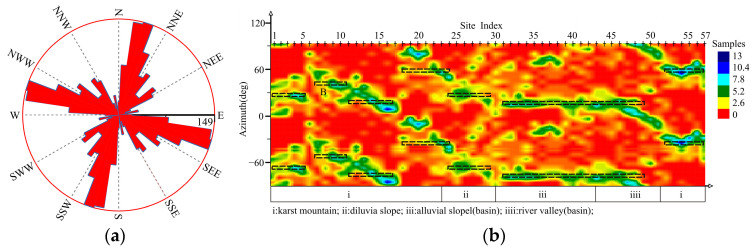
Impedance tensor decomposition of MT01: (**a**) rose diagram (**b**) site-based cloud diagram.

**Figure 5 sensors-24-07001-f005:**
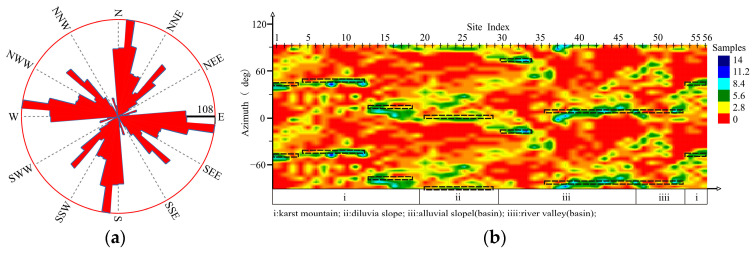
Impedance tensor decomposition of MT02: (**a**) rose diagram, (**b**) site-based cloud diagram.

**Figure 6 sensors-24-07001-f006:**
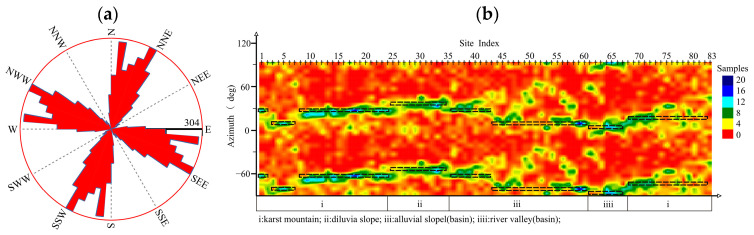
Impedance tensor decomposition of MT03: (**a**) rose diagram, (**b**) site-based cloud diagram.

**Figure 7 sensors-24-07001-f007:**
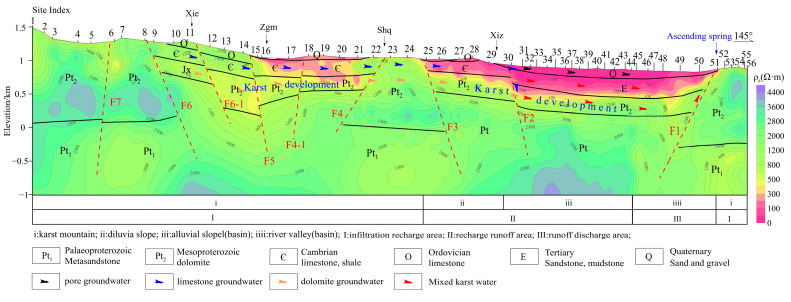
**Two-dimensional** inversion and interpretation results of MT01 profile.

**Figure 8 sensors-24-07001-f008:**
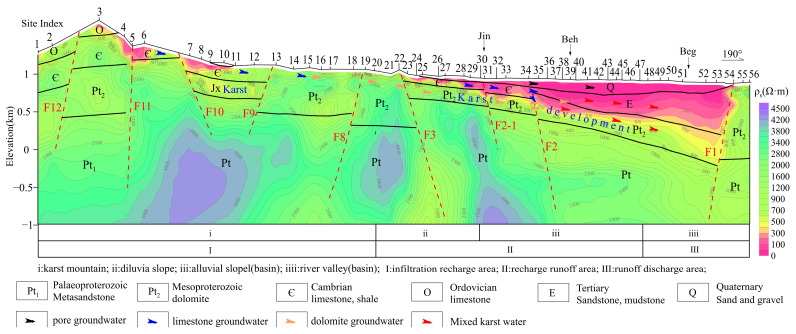
**Two-dimensional** inversion and interpretation results of MT02 profile.

**Figure 9 sensors-24-07001-f009:**
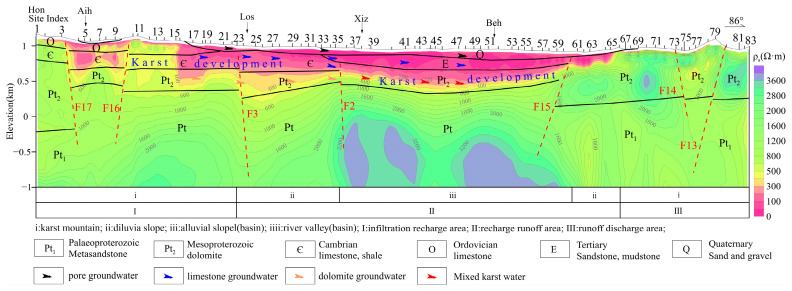
**Two-dimensional** inversion and interpretation results of MT03 profile.

**Table 1 sensors-24-07001-t001:** Geo-electrical strikes of MT01.

Group Index	Site Index	Geo-Electrical Strikes	Geo-Electrical Type
A	1–5	29°	Karst mountain
B	6–11	41°	Karst mountain
C	12–17	20°	Karst mountain
D	18–23	59°	Karst mountain
E	24–29	30°	Diluvia slope
F	30–50	15°	Alluvial slope
G	51–57	59°	Karst mountain

**Table 2 sensors-24-07001-t002:** Geo-electrical strikes of MT02.

Group Index	Site Index	Geo-Electrical Strikes	Geo-Electrical Type
A	1–2	41°	Karst mountain
B	3–12	45°	Karst mountain
C	13–19	15°	Karst mountain
D	20–29	5°	Diluvia slope
E	30–34	72°	Alluvial slope
F	35–53	9°	Alluvial slope
G	54–56	42°	Karst mountain

**Table 3 sensors-24-07001-t003:** Geo-electrical strikes of MT03.

Group Index	Site Index	Geo-Electrical Strikes	Geo-Electrical Type
A	1–2	29°	Karst mountain
B	3–8	12°	Karst mountain
C	9–23	28°	Karst mountain
D	24–35	30°	Diluvia slope
E	36–43	28°	Alluvial slope
F	44–60	9°	Alluvial slope
G	61–68	5°	Diluvia slope
H	69–83	19°	Karst mountain

## Data Availability

The datasets used and/or analyzed during the current study are available from the authors upon reasonable request.

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
