# Peer review of "Isotopic and Geophysical Investigations of Groundwater in Laiyuan Basin, China"

_sensors, 2024, doi:10.3390/s24217001_

Round 1
Reviewer 1 Report
Comments and Suggestions for Authors
1. In the introduction section, although the article provides an overview of the background of the study, it is recommended that more specific geological and hydrogeological data be added, especially the latest research results directly related to the laiyuan Basin, so that the reader can understand the starting point and necessity of the study in greater depth.
2、The purpose of the study is clear, but the elaboration of the significance of the study, especially the potential contribution to regional water resource management and ecological protection, could be added to enhance the practical application value of the study.
3. It is recommended that the specific methods of isotope analysis (e.g. sample collection, processing, analysis process, etc.) and the specific steps of parameter setting, data collection and processing of geophysical exploration techniques be described in detail. At the same time, the software used, instrument model and calibration process should be clearly mentioned to improve the reproducibility of the method.
4. When describing the experimental design, in addition to the sampling points and time, the basis for the calculation of the sample size, the reasonableness of the spatial distribution, as well as the possible deviations and solutions should be explained in detail. In addition, environmental factors (e.g. seasonal changes, human activities, etc.) that may affect the results should also be discussed.
5. It is suggested that Figure 2 should be modified, and it is sufficient to retain (d) to illustrate the problem. In addition, adjust the picture clarity.
6. When analysing the results, the intrinsic connection between the isotope data and the geophysical sounding results should be explored in more depth, and how they can together reveal the groundwater migration and recharge mechanisms. At the same time, outliers or results that do not meet expectations should be discussed in detail and possible explanations or hypotheses proposed.
7. In the discussion section, a more systematic comparative analysis with previous studies should be made to clarify the innovations and differences of this study. At the same time, the insights and contributions of this study should be presented for the controversial or unresolved issues in previous studies.
8. The conclusion section should summarise the findings of the study more concisely and clearly indicate the specific contribution of the study to the field of groundwater science. At the same time, it may be appropriate to expand on the application of the conclusions with suggestions for future research or practice.
9. If a large amount of raw data or detailed experimental records were generated during the study, it is recommended that they be provided as supplementary materials to allow the reader to further understand the process and results of the study. These materials may include data tables, graphs, experimental photos or videos, etc.
10、References appear in Chinese, please carefully check the accuracy and completeness of the references to ensure that all cited documents have been listed correctly and arranged in a uniform format.
Comments on the Quality of English LanguagePlease ensure that the full text is linguistically fluent, logically clear, and in line with the norms of academic writing. Also, check and adjust the format of the article (e.g. font, line spacing, heading levels, etc.) to improve overall readability.
Author Response
Comments 1: In the introduction section, although the article provides an overview of the background of the study, it is recommended that more specific geological and hydrogeological data be added, especially the latest research results directly related to the laiyuan Basin, so that the reader can understand the starting point and necessity of the study in greater depth. |
Response 1: Thank you for pointing this out. We agree with this comment. Therefore, In the introduction section, we add the latest research results of the Laiyuan basin (line 51-70), and in the second chapter, we supplement the geological and hydrogeological data (line 107-115).
|
Comments 2: The purpose of the study is clear, but the elaboration of the significance of the study, especially the potential contribution to regional water resource management and ecological protection, could be added to enhance the practical application value of the study. |
Response 2: We agree with this comment. Therefore, In the conclusion section, we add the the potential contribution to regional water resource management and ecological protection (line 363-370).
Comments 3: It is recommended that the specific methods of isotope analysis (e.g. sample collection, processing, analysis process, etc.) and the specific steps of parameter setting, data collection and processing of geophysical exploration techniques be described in detail. At the same time, the software used, instrument model and calibration process should be clearly mentioned to improve the reproducibility of the method. Response 3: We agree with this comment. Therefore, In the third section, we have added the specific methods of isotope analysis and geophysical exploration technical parameter setting, data acquisition and processing, as well as the software and instrument models used. (line 129-166).
Comments 4: When describing the experimental design, in addition to the sampling points and time, the basis for the calculation of the sample size, the reasonableness of the spatial distribution, as well as the possible deviations and solutions should be explained in detail. In addition, environmental factors (e.g. seasonal changes, human activities, etc.) that may affect the results should also be discussed. Response 4: We agree with this comment. Therefore, In the third section, In the description of the experimental design, we detailed the seasonal variation factors that affect the stable isotope analysis test (line 129-132), avoid the human interference factors of MT data acquisition (line 157-159), and explain the rationality of spatial distribution and the basis of sample size (line 149-156).
Comments 5: It is suggested that Figure 2 should be modified, and it is sufficient to retain (d) to illustrate the problem. In addition, adjust the picture clarity. Response 5: We agree with this comment. Therefore, In Figure 2, only d is retained and the resolution is adjusted.
Comments 6: When analysing the results, the intrinsic connection between the isotope data and the geophysical sounding results should be explored in more depth, and how they can together reveal the groundwater migration and recharge mechanisms. At the same time, outliers or results that do not meet expectations should be discussed in detail and possible explanations or hypotheses proposed. Response 6: We agree with this comment. In analyzing the results, we delve deeper into the intrinsic links between isotopic data and geophysical detections (line 289-303, 314-324, 327-340).
Comments 7: In the discussion section, a more systematic comparative analysis with previous studies should be made to clarify the innovations and differences of this study. At the same time, the insights and contributions of this study should be presented for the controversial or unresolved issues in previous studies. Response 7: We agree with this comment. the geophysical results have described the aquifer communication and recharge and discharge channels inferred by stable isotope characteristics in detail, and achieved qualitative to quantitative characterization (line 336-340).
Comments 8: The conclusion section should summarise the findings of the study more concisely and clearly indicate the specific contribution of the study to the field of groundwater science. At the same time, it may be appropriate to expand on the application of the conclusions with suggestions for future research or practice. Response 8: The conclusion is summarized and simplified, and the specific contribution of the research to the field of groundwater science is clearly pointed out. (line 352-359).
Comments 9: If a large amount of raw data or detailed experimental records were generated during the study, it is recommended that they be provided as supplementary materials to allow the reader to further understand the process and results of the study. These materials may include data tables, graphs, experimental photos or videos, etc. Response 9: Unfortunately, we cannot provide the original data due to the confidentiality of the project.
Comments 10: References appear in Chinese, please carefully check the accuracy and completeness of the references to ensure that all cited documents have been listed correctly and arranged in a uniform format. Response 10: We agree with this comment. Therefore, In Figure 2, We have revised reference [20].
|

Reviewer 2 Report
Comments and Suggestions for Authors
In this study, the characteristics of hydrogen and oxygen isotopes were studied by means of geomagnetic impedance tensor decomposition and two-dimensional precise inversion techniques. The stratigraphic structure of Lianhua Basin has been carefully explored and the characteristics of groundwater migration and recharge have been comprehensively studied. It combines geochemical and geophysical methods. Firstly, the geomorphology was constructed by geochemical method, and then the structural characteristics of Laiyuan Basin were comprehensively characterized by geomagnetic method and inversion technology. The formation structure, fracture and karst fracture development law of each water-bearing rock mass were studied in detail. According to the experimental results, combining the two methods is effective, but there are some problems here.
1. The study only used geochemical methods to construct features for subsequent inversion, but are there other methods available for feature construction? What are the performance advantages of the features constructed by the methods in this article compared to other features? Performance comparison is also needed in the experiment.
2. In the introduction, the author argues that the geomagnetic method has advantages over other methods. However, in the experimental part, the author only used the geomagnetic method, hoping that the author can also add other methods in the experiment to prove this with experimental results. It can also prove whether the feature construction method proposed in this paper has better performance than all other methods.
The wording in the text required some changes.
Author Response
Comments 1: The study only used geochemical methods to construct features for subsequent inversion, but are there other methods available for feature construction? What are the performance advantages of the features constructed by the methods in this article compared to other features? Performance comparison is also needed in the experiment. |
Response 1: Thank you for pointing this out. There are no other methods available for feature construction in this paper. The advantage of our method is that it provides a detailed characterization of the communication and supply/discharge channels of the aquifer inferred from stable isotope features, achieving a qualitative to quantitative characterization.
|
Comments 2: In the introduction, the author argues that the geomagnetic method has advantages over other methods. However, in the experimental part, the author only used the geomagnetic method, hoping that the author can also add other methods in the experiment to prove this with experimental results. It can also prove whether the feature construction method proposed in this paper has better performance than all other methods. |
Response 2: The interpretation depth requirement for this article is 2000 meters, which cannot be met by other electromagnetic methods. Therefore, we only chose the magnetotelluric method for this paper. |

Reviewer 3 Report
Comments and Suggestions for Authors
The following questions need to be addressed before considering for publication:
1) In Figure 1, are the words "tuyu syncine" actually "tuyu syncline"?
2) Section 3.1 lacks details about the specific steps and conditions under which the analysis was performed, making it hard for others to replicate the study.
3) In Figure 2, the δD-δ18O relationship was plotted from test results of 51 groups of samples, but the figure lacks error bars to indicate measurement variability.
Author Response
Comments 1: In Figure 1, are the words "tuyu syncine" actually "tuyu syncline"? |
Response 1: Thank you for pointing this out. We agree with this comment. Therefore, We have changed 'tuyu syncine' to 'tuyu syncline' (Figure 1).
|
Comments 2: Section 3.1 lacks details about the specific steps and conditions under which the analysis was performed, making it hard for others to replicate the study. |
Response 2: We agree with this comment. Therefore, In the section 3.1 and 3.2, we have added the specific methods of isotope analysis and geophysical exploration technical parameter setting, data acquisition and processing, as well as the software and instrument models used. (line 129-166).
Comments 3: In Figure 2, the δD-δ18O relationship was plotted from test results of 51 groups of samples, but the figure lacks error bars to indicate measurement variability. Response 3: We agree with this comment, However , the δD was determined by high temperature thermal conversion elite-isotope ratio mass spectrometry (HTC-IRMS) with an accuracy of 1.0‰. δ18O was determined by Gasbench II isotope ratio mass spectrometry (Gasbench II -IRMS) with an accuracy of 0.2‰. Therefore, the error bars of δD and δ18O values are very small and cannot be clearly displayed when added to the graph. |

Reviewer 4 Report
Comments and Suggestions for Authors
This is a very interesting topic of hydrogeological interpretation by combining HO isotopic and magnetotelluric data. The structure is good, the length is a little short as article type publication. A few detailed comments and suggestions listed might be helpful for author to improve this manuscript to final publication.
1. line 98: authors could add more details of materials and methodology.
2. Figure 2: add the water sample ID in the plot for comparing and analysis, or a figure to show the ratio in map for interpretation easier.
3. line 239: more details needed about 2D inversion parameters, e.g. cell size, initial resistivity model etc.; wonder if author did compare with 3D inversion results.
4. line 192,246: About 'L curve analysis method', 'conjugate impedance method', authors can add description or references; and which inversion software/package is used to calculate subsurface resistivity?
5. line 252: might be re-edit the text 'At the fault F2, the Cambrian strata began....''which is inferred to be the groundwater runoff channel...', the sentences repeated two times in line 264 and line 274..
6. line 268: author can create a 3D cross-plot subsurface interpretation to display and compare? edit the color bar with same scale and interval for comparison in Figure 7,8,9.
7. Figure 9: looks there is a fault at MT site #15? wondering what makes this variation.
8. Based on the topography in Figure 7, 8, 9, it might be reasonable that the flow direction of groundwater is from left to right(W-E) in Figure 9. Wonder if there is any other identification since the interpreted dolomite depth is different and the water spring is close to the end of line MT01. Please state.
9. line 242: TM polarization mode is less affected by 3D effect, could add TE mode as input for combined inversion.?
10. adding interpreted layers 'black lines and fault dashed lines' in legend; two MT site 31 in Figure 7..
11. Author can read through more times to reduce language error, typos. If author used a translation software, please re-edit and polish the text, such as line 117-120.
Author Response
Comments 1: line 98: authors could add more details of materials and methodology. |
Response 1: Thank you for pointing this out. We agree with this comment. Therefore, In the section 3.1 and 3.2, we have added the specific methods of isotope analysis and geophysical exploration technical parameter setting, data acquisition and processing, as well as the software and instrument models used. (line 129-166).
|
Comments 2: Figure 2: add the water sample ID in the plot for comparing and analysis, or a figure to show the ratio in map for interpretation easier. |
Response 2: We agree with this comment. Therefore, In the conclusion section, we have added the water sample ID in the plot for comparing and analysis (Figure 2).
Comments 3: line 239: more details needed about 2D inversion parameters, e.g. cell size, initial resistivity model etc.; wonder if author did compare with 3D inversion results. Response 3: We agree with this comment. The one-dimensional Bostick inversion results of statically corrected TE model are used as the initial resistivity model. The inversion cell size increases in depth by 10 meters. (line 289-293). We did not compare the results with 3D inversion results.
Comments 4: line 192,246: About 'L curve analysis method', 'conjugate impedance method', authors can add description or references; and which inversion software/package is used to calculate subsurface resistivity? Response 4: We agree with this comment. Therefore, we have added references for 'L curve analysis method', 'conjugate impedance method'(line 237), and MT-Pioneer software is used to calculate subsurface resistivity. (line 293).
Comments 5: line 252: might be re-edit the text 'At the fault F2, the Cambrian strata began....''which is inferred to be the groundwater runoff channel...', the sentences repeated two times in line 264 and line 274. Response 5: We agree with this comment. Therefore, we re-edit the text 'At the fault F2, the Cambrian strata began....''which is inferred to be the groundwater runoff channel...'(line323-324, 336-340).
Comments 6: line 268: author can create a 3D cross-plot subsurface interpretation to display and compare? edit the color bar with same scale and interval for comparison in Figure 7,8,9. Response 6: Due to the large difference in the elevation of each Section line, it is impossible to add terrain to the 3D cross map, which will affect the comparison of the map after interpolation. So we don't recommend drawing a 3D cross plot, but we have the capability to do it.
Comments 7: Figure 9: looks there is a fault at MT site #15? wondering what makes this variation. Response 7: We analyzed that this is not a fracture. Because the resistivity difference here is mainly caused by the difference in the degree of karst development. The degree of karst development is low, the water content is small, and the resistivity is high. The degree of karst development is high, the water content is large, and the resistivity is low.
Comments 8: Based on the topography in Figure 7, 8, 9, it might be reasonable that the flow direction of groundwater is from left to right(W-E) in Figure 9. Wonder if there is any other identification since the interpreted dolomite depth is different and the water spring is close to the end of line MT01. Please state. Response 8: There is currently no other identification available.
Comments 9: line 242: TM polarization mode is less affected by 3D effect, could add TE mode as input for combined inversion.? Response 9: The one-dimensional Bostick inversion results of TE model are used as the initial resistivity model for inversion (line289-291).
Comments 10: adding interpreted layers 'black lines and fault dashed lines' in legend; two MT site 31 in Figure 7. Response 10: we have added the interpreted layers 'black lines and fault dashed lines' in legend (Figure 7,8,9), and Modified two MT sites 31 in Figure 7. Comments 11: Author can read through more times to reduce language error, typos. If author used a translation software, please re-edit and polish the text, such as line 117-120. Response 11: we have re-edit and polish the text (in red). |

Round 2
Reviewer 2 Report
Comments and Suggestions for Authors
I have no more comments.